# Machine Learning Approaches for the Prioritization of Genomic Variants Impacting Pre-mRNA Splicing

**DOI:** 10.3390/cells8121513

**Published:** 2019-11-26

**Authors:** Charlie F Rowlands, Diana Baralle, Jamie M Ellingford

**Affiliations:** 1North West Genomic Laboratory Hub, Manchester Centre for Genomic Medicine, Manchester University Hospitals NHS Foundation Trust, St Mary’s Hospital, Manchester M13 9WJ, UK; charles.rowlands@manchester.ac.uk; 2Division of Evolution and Genomic Sciences, School of Biological Sciences, Faculty of Biology, Medicine and Health, University of Manchester, Manchester M13 9PR, UK; 3Human Development and Health, Faculty of Medicine, University of Southampton, MP808, Tremona Road, Southampton SO16 6YD, UK

**Keywords:** Mendelian disease, diagnostics, variant interpretation, variant prioritization, RNA splicing, bioinformatics, machine learning, genomic medicine, effect prediction

## Abstract

Defects in pre-mRNA splicing are frequently a cause of Mendelian disease. Despite the advent of next-generation sequencing, allowing a deeper insight into a patient’s variant landscape, the ability to characterize variants causing splicing defects has not progressed with the same speed. To address this, recent years have seen a sharp spike in the number of splice prediction tools leveraging machine learning approaches, leaving clinical geneticists with a plethora of choices for in silico analysis. In this review, some basic principles of machine learning are introduced in the context of genomics and splicing analysis. A critical comparative approach is then used to describe seven recent machine learning-based splice prediction tools, revealing highly diverse approaches and common caveats. We find that, although great progress has been made in producing specific and sensitive tools, there is still much scope for personalized approaches to prediction of variant impact on splicing. Such approaches may increase diagnostic yields and underpin improvements to patient care.

## 1. Introduction

Accurate interpretation of genomic variants is the backbone of genomic medicine. Determining the causative variant in patients with Mendelian disorders facilitates both management and potential downstream treatment of the patient’s condition, as well as providing peace of mind and allowing more effective counselling for the wider family. Rare genomic disorders impact 1 in 17 individuals [1], can be severely debilitating or life-limiting, and may also require expensive specialist care, placing a huge emotional burden on sufferers and their families. Thus, there is a moral and economic imperative to ensure as many patients as possible receive molecular diagnoses, and to ensure there is clarity in diagnostic interpretation of genomic variation. This is further emphasized by the recent revolution of genomic sequencing within healthcare in the UK, where diverse diagnostic testing approaches are available nationwide for many disease subtypes [2].

The advent of next generation sequencing (NGS) technologies, which allow accurate, high-throughput identification of the body of variants in an individual’s genome, has revolutionized the way we generate genomic patient data. Gene panels—sequencing workflows designed to identify variants in a subset of disease-related genes—have shown great promise for improving diagnostic rates in diverse disease areas, for example, inherited retinal dystrophies [3] and hereditary tumor syndromes [4]. Exome and genome sequencing are now also commonplace in the diagnosis of Mendelian patients and allow identification of the vast majority of coding variants in a patient’s genome [5,6,7,8].

Despite these recent improvements in the ease with which we generate genomic datasets, certain causative mutations still often elude diagnostic pipelines, and, as such, molecular diagnosis rates remain between 25–40% for most diagnostic centers [9,10]. Large and/or complex structural genomic rearrangements, for example, are often difficult to identify from short-read sequencing alone, although strategies to incorporate copy number variations are now commonplace in diagnostic settings and have been shown to increase diagnostic yields [11,12]. Furthermore, pathogenic variation in non-coding regions, which may cause disease through impairing RNA secondary structure [13] or gene expression and regulation [14,15], is difficult to discern from genomic datasets alone.

One similarly underappreciated source of pathogenic variation is that impacting RNA splicing, the process by which introns are removed co-transcriptionally from nascent pre-mRNA transcripts. Recent estimates suggest between 9–30% of causative variants in Mendelian disorders cases may act through disruption of splicing [16], and so deeper consideration of variant impact on splicing is likely to be of great benefit in improving diagnostic yield for rare disease patients.

Recent research has shown that direct analysis of RNA mis-splicing via RNA-seq may prove highly beneficial in improving molecular diagnosis for disease subtypes such as neuromuscular disease [17,18]. However, the tissue-specific nature of splicing imposes limits on the subtypes of disease for which biopsy and subsequent RNA-seq analysis of relevant tissues may be effective.

Recent years have seen a surge in bioinformatics tools designed to predict variant impact on splicing, and these offer an opportunity to circumvent many limitations of RNA-seq-based approaches. An increasing number of these tools rely on machine learning—computational approaches that can identify patterns in data and use this knowledge to speculate on new data. Machine learning is already being applied to great effect in diverse biological fields, such as the modelling of social networks in animal behavior studies [19,20] and protein secondary structure prediction [21]. The application of machine learning to the prediction of variant impact on splicing has been assisted by the recent availability of large-scale transcriptomic datasets, such as the GTEx project [22], which allow researchers to link genomic and transcriptomic variation across large numbers of individuals and tissue types [23,24,25].

Depending on sequencing strategies, clinical scientists will be expected to interpret and triage hundreds to millions of genomic variants per individual, although many variants can be immediately excluded due to their frequency in the general population [26]. The development of effective machine learning tools for the prediction of splicing impact will allow prioritization of likely pathogenic variants among the mass of genomic variants returned by standard diagnostic pipelines. Ultimately, these tools may prove a valuable asset in improving diagnostic yield globally.

This review aims to provide a summary of some of the major machine learning-based splice analysis tools released to date. While the focus here is largely on the functionality of these tools, some basics of machine learning are introduced to allow easier understanding of their computational basis, as described below. For a more thorough grounding in machine learning, the reader is pointed to an excellent recent review by Eraslan et al. [27], describing many aspects of computational modelling in genomics.

## 2. Pre-mRNA Splicing and Its Role in Pathogenesis

The co-transcriptional splicing of pre-mRNA is an intricate and tightly regulated process. Over the course of the splicing reaction, highly specific and complex interactions between the *trans*-acting components of the spliceosome and *cis*-acting sequences in the RNA transcript occur in a stepwise fashion. Mutations in the core spliceosome components themselves, as well as in auxiliary splice factors, can be causes of Mendelian disease. For example, mutations of the spliceosomal U5 snRNP component EFTUD2 cause the autosomal dominant craniofacial disorder mandibulofacial dysostosis, Guion–Almeda type [28], and mutations in several pre-mRNA processing factors are known as a cause of autosomal dominant inherited retinal disease, for example *PRPF31* [29] and *SNRNP200* [30].

Moreover, homozygous knockout of most splicing factors is embryonic lethal [31], demonstrating the vital importance of the splicing process in sustaining eukaryotic life.

However, splicing dysregulation in Mendelian disease predominantly occurs at the *cis*-acting level, where disruption of vital sequence elements leads to aberrant splicing events (Figure 1), such as single or multi-exon skipping [32,33] (Figure 1b), and the creation of new “cryptic” splice sites [34,35] (Figure 1c,d) which can cause the retention of whole introns [36,37] (Figure 1e) and the inclusion of pseudoexons [38,39,40] (Figure 1f). Besides the well-characterized mutation of canonical splice sites, disruption of all manner of *cis*-acting elements has been described as a cause of Mendelian disease; mutation of the wider splice region around canonical sites [41,42], exonic and intronic silencers/enhancers [43,44], the 3′ branch point [45,46], and poly-pyrimidine tract [47,48] have all been implicated in monogenic disease. As a result of the varied molecular mechanisms underlying splicing aberration, there are diverse sets of computational models for splicing prediction, including both those which focus on prediction of a particular subset of splice aberration, and others which adopt more holistic approaches to consider multiple possible impacts on splicing.

## 3. Early Computational Methodologies to Prioritize Genomic Variants Impacting Splicing

Despite the relatively recent advent of machine learning-based splicing models, many other computational approaches to the prediction of splice disruption have been described over the last two decades.

Many early tools for the prediction of splicing regulatory element (SRE) binding sites were based on position weight matrices (PWMs)—log-scaled representations of the frequency of particular nucleotides within sequences predicted to bind splicing factors. Experimental derivation of such PWMs formed the basis of tools such as ESEFinder [49] and Human Splicing Finder [50], and decreased fitting of mutant sequences to the PWM model was seen as evidence for impairment of splice factor binding.

Many computational and experimental approaches to splice prediction have involved the investigation of nucleotide hexamers (i.e., sequences of 6 bases length). The method RESCUE-ESE, for example, computationally identified 10 splice-enhancing hexanucleotides in the vicinities of weak splice sites [51]. An approach named ESRseq [52] made use of a saturation technique in which all 4096 possible nucleotide hexamers were scored for splicing impact based on in vitro minigene splicing assays. The tabulated results of these experiments, published online, could then be used to speculate on the splicing potential of a mutant sequence versus its wild-type counterpart.

One tool that remains widely used in splice site prediction is MaxEntScan [53]. Based on the principles of maximum entropy modelling (MEM) from the field of information theory, MaxEntScan generates two models based on a set of real and decoy splice sites. It then compares the probability that a presented nucleotide sequence belongs to each of the two distributions and returns how much more likely it is that the sequence is a real, rather than decoy, site.

## 4. Basics of Machine Learning Techniques

Splice prediction tools have incorporated a wide range of machine learning-based models. In-depth descriptions of deep learning architectures have been given extensively elsewhere [27]. Here, we instead give a broader overview of machine learning and cover basic elements of machine learning in the context of splicing analysis. Italicized terms are defined in Table 1.

### 4.1. Basics of Machine Learning

All machine learning models require both training and testing—to do this, a relevant data set is divided into both a training set and a test set. Importantly, no entry in one set is present in the other; were there to be overlap, the model would be over-trained to recognize those items in the test set that it had already seen, and measurement of model efficacy would show it to be more effective than it really was. The variables or characteristics in each dataset that are input to a model are termed features. Initially, some model-specific algorithm is applied (usually iteratively) to this training set to develop an initial model. The model is then applied to the test set and its efficacy quantified. Measurement across different versions of the model then allows the model to be fine-tuned to maximize its efficacy.

The efficacy of a machine learning model is generally measured as some kind of loss function – in essence, a measurement of how far a model’s predictions deviate from the expected outcome, and machine learning algorithms strive to minimize this value over the course of the generation of the model. In other words, these models are gradually tweaked so that their ability to accurately classify data improves over the training process.

### 4.2. Features

A key element of machine learning is the use of features: these are the underlying characteristics or variables that are input to the models and from which inferences are ultimately made. It is these features by which data are classified or separated. In the context of genomic and transcriptomic analysis, many of these features are often sequence-based, representing the frequency or position of particular nucleotide sequences over a given region. Biochemical features, such as GC content and thermodynamic properties, are often also employed. Moreover, some tools adopt a meta-analytical approach through the incorporation of output from other tools as features, such as the use of SPANR [54] and CADD [55] scores in S-CAP [56] (see below). Differences in choice of features may often underlie the various strengths and caveats of particular tools.

### 4.3. Training and Test Sets

One major contributing factor to the rapid surge in the number of machine learning-based splice prediction tools is the increased availability of publicly-available datasets. Particularly valuable are experimentally-derived RNA-seq datasets, which allow effective linking of genome- and transcriptome-level features. Several tools also incorporate measurements of pathogenicity in the form of variant classifications from ClinVar [57]. Many tools use raw sequence data as input; in such cases, these sequences are taken from a reputed transcript model, most often GENCODE [58], as in the cases of MMSplice [59] and SpliceAI [60].

### 4.4. Outputs

Machine learning models broadly fall into the categories of *regression* and *classification* models. Classification models identify the class (of a set of classes) to which an unseen data entry is most likely to belong. On the other hand, regression models use input data to predict a quantitative value. Thus, the output of a model depends on its design, the types of features which are utilized as input and the objectives of the prediction tool. Most splice prediction tools utilize regression models, and generate predictive scores corresponding to, for example, the strength of a novel splice site (SpliceAI), the magnitude of an exon skipping event (SPIDEX), or variant pathogenicity (S-CAP). How scores from these tools are utilized and interpreted is thus highly dependent on the tool being used.

### 4.5. Model Evaluation

As described above, many machine learning models refine themselves over training iterations by minimizing some kind of loss function. However, comparative analysis of the relative performance of different models usually relies on the construction of an unseen test dataset that can be applied to both/all models. Model performance metrics, such as the area under curve (AUC) of both receiver-operating characteristic (ROC) and precision-recall (PR) curves, can then be used to more directly compare model performance, although this may be confounded by many factors (see Discussion).

## 5. Common Machine Learning Models in Splice Prediction

In this section, we present a non-exhaustive list of some common machine learning architectures seen in splice prediction software.

### 5.1. Support Vector Machines (SVMs)

SVM models aim to use a *hyperplane* (a surface with one fewer dimension than the space around it) to separate data belonging to different classes. This is done such that the distance between the hyperplane and data that lie closest to the overlap between two classes—the so-called *support vectors* —is maximized (Figure 2a). Data presented to an SVM are then classified according to which side of the hyperplane they lie on. *Multiclass SVM* approaches can also be used where there are more than two outcome classes to which data may be assigned. Finally, data which cannot be separated by a single continuous hyperplane (Figure 2b) are able to be transformed using the *kernel trick*. This approach makes use of kernel functions—mathematical operations that allow inference of relational qualities between data points in a computationally inexpensive manner. Common kernels used in machine learning are the polynomial and radial basis function (RBF) kernels, although a multitude of others exist.

Importantly, standard SVMs are only able to classify data as belonging to one group or another; to provide probabilistic measures of confidence or effect size, models need to be adapted and extended.

### 5.2. Decision Trees

Decision trees are a simple but powerful form of machine learning model in which a series of binary choices is designed that produces the most effective classification or prediction of a dependent variable (Figure 2c)—this is done through selecting whichever choice allows most accurate separation of data at each stage in the tree-building process. The single decision tree that is generated for a given training set, however, is prone to overfitting and bias for the input data. To remedy this, random forest models are often used (Figure 2d). Here, iterative *bagging* (bootstrap aggregating) of the training data, as well as of the variables considered at each stage of the tree-building process, allows the model to be more generalizable to unseen data. Gradient tree boosting (Figure 2e) adopts a different approach to bypass overfitting by using the generation of successive trees, each of small contribution to the final model, until decreases in the model loss are negligible.

### 5.3. Deep Neural Networks (DNNs) 

DNNs are computational networks modelled on the activity of biological *neurons* (Figure 2f). These neurons are arranged in layers (Figure 2g): the first is an input layer, where each neuron is assigned a value corresponding to a feature of the model for that data entry. The final layer contains neurons corresponding to the possible outcomes of the model. Between these are a number of *hidden layers*. Hidden layer neurons receive weighted input from all the neurons in the previous layer, and subsequently distribute the sum of these inputs to all neurons in the next layer by another series of weighted connections. These weightings are assigned at random before the training of the model.

Training data are presented sequentially to a DNN and the resulting output in the final layer recorded and averaged over many training iterations. The efficacy of the model is then compared in relation to expected results. Through a process termed *backpropagation*, the weightings of the connections between neurons are proportionally adjusted so as to minimize the loss function of the model. This is repeated over multiple presentations of training data, or *epochs*, gradually refining the model. Particularly popular in the analysis of nucleotide sequences is a variation termed the convolutional neural network, or CNN, in which input data are ordered in the form of an *n*-dimensional array—that is, nucleotides are input to the model in windows.

## 6. Machine Learning-Based Tools for Splicing Prediction

Below, we describe 7 tools incorporating different aspects of splicing prediction. We provide a tabulated summary of key characteristics of each model for reference (Table 2), with more information on code/score access given in Appendix A. We also include a tabulated and schematic representation of the transcript regions amenable to analysis by each tool, using the pre-mRNA transcript of the APO3 gene as an exemplar (Table 3, Figure 3).

### 6.1. CADD (Combined Annotation-Dependent Depletion)

CADD [55,61] was among the earliest machine learning-based variant scoring systems; it generates a score that is approximately interpretable as a measure of pathogenicity.

To train the CADD model, both benign and pathogenic variant sets were derived. For the former, variants with high mean allele frequency (≥95%) in the 1000 Genomes dataset [67] were chosen that had arisen since the split between humans and chimpanzees, based on the assumption that such variants had been fixed under natural selection, and so are, at worst, weakly pathogenic. De novo pathogenic variants—both indels and SNVs—were simulated genome-wide using a model informed by local mutation rates and CpG dinucleotide mutation asymmetry.

A wide range of features were incorporated into the CADD model. Such features included: conservation metrics, such as phyloP [68], GERP [69], and phastCons [70]; regulatory information, such as transcription factor binding [71] and DNAse I hypersensitivity regions [72]; and protein-level predictions, for example Grantham [73], SIFT [74], and PolyPhen [75] scores. Transcript-level features, such as gene expression levels, were also derived, along with some consideration of splicing in the inclusion of variant distance to the nearest canonical splice site.

The initial releases of CADD adopted an SVM-based approach (Figure 1) with a linear kernel. However, with later releases, L_2_-regularized logistic regression—a form of regression model allowing the modeling and prediction of a binary dependent variable—was shown to lead to improved sensitivity and specificity, and so became the model of choice [55].

CADD has been rapidly and widely adopted since its creation, with uses in pathogenicity prediction for many disease subtypes, both Mendelian and complex. In a study of autism spectrum disorder (ASD) in 85 quartet families, for example, CADD scoring was used to filter genomic variants of interest, resulting in the identification of ASD-relevant mutations in 69.4% of affected siblings [76].

The use of CADD scoring has become a gold standard for the prediction of protein-coding variant impact. This ubiquity has led to CADD becoming a benchmark against which many predictive tools are measured. However, its efficacy in terms of splicing prediction is undermined by certain features: the use of conservation scores, for example, may not be informative at the poorly-conserved bases of introns, where cryptic splice sites and pseudoexonisation events are liable to occur. Thus, while a highly effective tool for protein-coding impact prediction, CADD lacks the splice-specific considerations to accurately predict variant effect at the transcript level.

### 6.2. TraP (Transcript-Inferred Pathogenicity) Scores

TraP [62] is a random forest-based tool (Figure 2d) for the analysis of non-coding variant impact at the transcript level, providing a score between 0–1 to reflect the scale of this impact. This score corresponds to the proportion of decision trees in the model that predict a variant as pathogenic, and may thus be used as a proxy for the degree of impact a variant is likely to have on a transcript.

TraP was trained on 75 pathogenic and 402 benign variants. To source the former, the authors curated a list of solely synonymous variants associated with rare disease to avoid any incorporation of protein-coding consideration in the model. Synonymous de novo variants in healthy individuals were selected as the benign dataset. These rare variants were selected over common variants in the population to avoid training the model to distinguish solely between rare and common variants.

The TraP model consists of 20 features, primarily splicing-related, including whether or not the variant lies within the splice site region (as pre-defined by the authors); the score of new splice sites where cryptic GT-AG dinucleotides are introduced, according to a position-specific scoring matrix (PSSM); and a bespoke “variant regulatory score”, which incorporates several other features that do not directly affect existing splice sites. The model further incorporates the GERP++ conservation metric [77]. The random forest model underlying TraP consists of 1000 decision trees harboring various combinations of these 20 features.

The authors suggest a 3-tier threshold system for TraP scoring: variants with a TraP score below 0.495 are considered likely benign. Variants scoring ≥0.495 but below 0.93 are in an intermediate range, representing variants that may possibly have an impact at the transcript level. Variants scoring ≥0.93 are likely pathogenic. When considering intronic variants, the authors suggest a threshold of 0.75 to avoid inclusion of large numbers of false positives.

The authors compared the performance of TraP compared to CADD in distinguishing pathogenic and benign variants, both intronic and synonymous. They demonstrated that matching the specificity of TraP at a 0.495 threshold would give CADD a sensitivity of just 6% or 18.8%, for synonymous and intronic variants, respectively. Thus, it is evident that TraP scoring offers a marked improvement on the CADD model for the prioritization of variants impacting splicing.

In addition, TraP considers the potential impact of variants across multiple transcripts, a feature not considered by many splicing prediction tools. The efficacy of the model is also impressive, particularly given the relatively small size of the training and test sets. While the model works well in identifying pathogenic intronic variants, retraining a second model using such pathogenic intronic variants, rather than synonymous ones, may improve the performance of TraP yet further.

### 6.3. SPANR (Splicing-Based Analysis of Variants)

SPANR [54] seeks to model variants impacting cassette exon splicing—the inclusion or skipping of a given internal exon—across a number of human tissues. It achieves this using a Bayesian deep learning model based on the percentage spliced in (PSI, or Ψ) metric, a measure of the percentage of mature mRNA transcripts containing, rather than excluding, a particular exon. This model seeks to maximize a “code quality” metric that is a measure of the improvement of the model to predict Ψ values over a random guesser. SPANR works on a variation of a two-layer neural network, where the hidden layers of the model are common to all tissues, but each tissue has a distinct output layer.

Transcripts from RefSeq [78] were mined, and Human UniGene data from NCBI analyzed, to identify instances of cassette and constitutive exon splicing in the normal human transcriptome, leading to the identification of 10,689 cassette and 33,159 constitutive exons (all flanked by an exon on either side). The Ψ metrics for each of these central exons was then computed genome-wide using RNA-seq data from the Illumina BodyMap 2.0 project (NCBI GSE30611) and used as input for training an ensemble of DNN models. Δψ values representing the predicted change in exon inclusion were then able to be generated, with the paper using |Δψ ≥ 5%| as a general threshold over which a variant is considered to impact cassette exon splicing.

In the original paper, the authors demonstrated the utility of SPANR in the analysis of specific variant cohorts in patients with spinal muscular atrophy (SMA) and Lynch syndrome, implicating common causative variants in these disorders as splice-impacting. They also showed that predicted effects of simulated variants in intron 7 of the SMN2 gene are recapitulated with RT-PCR. They conducted a wider analysis of SNVs in genome data from 5 patients with autistic spectrum disorder and observed an enrichment of splice-impacting variants in genes associated with neurodevelopmental roles, thus demonstrating a wide range of potential uses for the tool in the study of both Mendelian and complex disease.

The model is somewhat limited by the scope of the cassette exon model—a variant must lie within 300 bp of an exon that itself lies between two other exons, meaning variants in first or terminal exons are not analyzable. This also renders the model obsolete for analysis of pathogenic variant types such as cryptic splice sites and deep intronic mutations. However, a webserver is provided, allowing easy analysis of small batches of variants, while a tabulated version of the SPANR dataset called SPIDEX, comprising pre-computed scores for all eligible variants in the genome, can be downloaded by the user and used during variant annotation with the ANNOVAR package for larger variant sets [79]. SPANR may thus be a powerful component of a predictive pipeline, but is likely too limited in scope to be considered proof of pathogenicity in isolation.

### 6.4. CryptSplice

CryptSplice [63] aims to predict the effects of the generation of cryptic splice sites. Namely, it considers three scenarios: the weakening of a canonical site by the introduction of a new splice site nearby, the outcompeting of a canonical site by a novel site, and the introduction of a functional deep intronic splice site.

An SVM forms the basis of CryptSplice, with input data being transformed with an RBF kernel, which was shown to yield the greatest accuracy. To provide probabilistic estimates to accompany classifications, the model was trained using 10-fold cross validation; that is, the training set was randomly divided into 10 equal parts and each part successively used to generate a new model. The distribution of accuracies across different models then formed the basis of probability metrics.

For training, CryptSplice was trained on a series of “true” splice sites derived from the NN269 [80] and HS^3^D [81] datasets, repositories of splice junctions curated from GenBank annotations following various quality control and cleaning processes. An equal number of “false” sites were derived, consisting of sequences with GT or AG dinucleotides at least 60 bp from a canonical splice site. All features for the model were sequence-based and fell into one of three categories (Table 2).

If a cryptic donor or acceptor is created, CryptSplice is able to cover regions >100 bp into the intron (Figure 3), lending it some strength over some tools that lack applicability far from splice junctions. However, some weaknesses in the model are apparent—the training junctions, for example, are derived from transcript annotations over 17 years old. Thus, the model may be underpowered to detect weaker splice sites that may not have become part of standard transcript models until more recently, and other tools are likely more effective for analysis of variants lying outside deeply intronic regions.

### 6.5. MMSplice (Modular Modeling of Splicing)

The tool MMSplice [59] aims to model the competitive interaction between splice sites in close proximity, supplementing this with predictions of exon skipping, splicing efficiency (i.e., the proportion of transcripts undergoing, rather than bypassing, splicing at a particular junction) and pathogenicity.

MMSplice has a complex underlying modular architecture containing 6 basic models of the transcript space (Figure 4a), covering donor and acceptor sites, plus 3′ and 5′ intronic and exonic sequences. Each was generated by a neural network with 2–4 layers, and all but the donor model had at least one convolutional layer. To generate the donor and acceptor models, all splice donor and acceptor sites present in the GENCODE v24 annotation [58] were derived as examples of positive sites. Random sequences from within the same genes were then used as negative sequences, provided they did not overlap the position of the positive splice sites. The output of these models is a positive or negative score, corresponding approximately to the strength of the presented variant sequence as a donor/acceptor.

To generate the 5′ and 3′ exonic and intronic models, the authors leveraged a massively parallel reporter assay (MPRA) generated by Rosenberg et al. [64], in which the relative splicing efficiencies of pairs of random 25-mer oligonucleotides were evaluated on both the exonic and intronic sides of an intronic splice junction. These models derive either Δψ_5_ or Δψ_3_ metrics, corresponding to the relative usage of a variant sequence as a splice acceptor or donor, respectively, compared to the canonical site.

A series of regression models were then designed based on the output of these models in order to predict variant impact on splicing. Four linear regression models were constructed: one analyzed variant impact on exon skipping through analysis of data from the splice analysis pipeline Vex-seq [82]; two were designed to predict Δψ_5_ or Δψ_3_ values based on cross-referencing of genome and RNA-seq data from the GTEx study [22]; the fourth leveraged a massively parallel splicing assay, or MaPSy [83], to predict splicing efficiency. In addition to this, a logistic regression model to predict pathogenicity was derived based on known pathogenic and benign variants in the splice region, as listed on ClinVar [57]. Thus, MMSplice provides a powerful combination of both biological and clinical predictions.

MMSplice is highly intricate and versatile, and is also easily clinically applicable, being able to take variant call format (VCF) files as input, and incorporating both SNV and indel predictions (unlike many tools) to predict a wide range of variant impacts on splicing. However, the training set of all splice junctions in the GENCODE v24 annotation may also contain substantial numbers of false positives where particular transcripts have been computationally predicted and remain experimentally unverified. Furthermore, modelling of competitive splice site interactions using GTEx data was based solely on samples from brain and skin tissue, which may underpower the model for predicting competitive interactions that predominate in other tissue types.

### 6.6. S-CAP (Splicing Clinically Applicable Pathogenicity Prediction)

S-CAP [56] is a splice prediction tool designed to directly predict the pathogenicity of splice-impacting variants. Much like MMSplice, S-CAP compartmentalizes the splicing landscape. In S-CAP, this compartmentalization comprises 6 distinct regions: 3′ intronic, 3′ core, exonic, 5′ core, 5′ extended, and 5′ intronic (Figure 4b), all lying within 50 bases of the canonical exon-intron junction. This approach aims to counter the tendency for prioritization of core splice site mutations in most machine learning models, which may understate the pathogenicity of more intronic variants.

S-CAP took both the Human Gene Mutation Database (HGMD) [84] and ClinVar [57] as sources for pathogenic variation, while benign variants were sourced from gnomAD (minor allele frequency ≥1%). The model is trained on 29 different features, classified as chromosome, gene, exon or variant level features. These features include highly tailored analyses, such as the number of rare variants found in the given exonic locus, or the SPANR and CADD scores for the variant. Intolerance of the gene as a whole to mutation is incorporated into the model through the use of pLI (probability of being loss-of-function intolerant) [65] and RVIS (residual variation intolerance) [66] scores.

In cases of 5′ and 3′ core mutations, the downstream consequence is almost universally impairment of splicing, removing the requirement of evaluating splice impact. This leaves only the question of whether this impairment of splicing is likely pathogenic. This is highly dependent on whether the variant is present in a heterozygous or hemi/homozygous state. To this end, core splice variants are run through two models, one based on a recessive and the other on a dominant inheritance model, and a score returned for each possibility.

Pre-computed scores are available for all variants lying within 1 of the 6 regions considered by the model, and individual thresholds are predefined for analysis of each of these regions. These thresholds, however, are designed for 95% sensitivity, coming somewhat at the expense of specificity and leading to the generation of large numbers of false positives also being identified. There is also substantial variety in the efficacy of the 6 models: exonic and 5′ intronic mutations are particularly difficult to characterize. This is most likely accounted for by the method of generation of these two models, for which variants had to be co-opted from other compartments prior to training, in order to boost an otherwise small pool of pathogenic variants. While S-CAP is underpowered to detect these variant types compared other types, it regardless outperformed SPIDEX, CADD and TraP in both sensitivity and specificity.

Although it doubtless plays a huge part in the efficacy of the tool, the division of the genomic landscape also comes at the expense of universal applicability: variants lying more than 50 bp into the intron are not covered by the model. Despite this, for the cohort of variants lying within these predefined regions, S-CAP has the potential to be a highly effective predictive tool.

### 6.7. SpliceAI

The deep learning tool SpliceAI [60] analyses each position in a pre-mRNA transcript and evaluates whether it is likely to be a splice donor, acceptor, or neither. The model considers all bases within 50 bp of a presented variant and returns the one with the most substantial gain or loss of acceptor or donor potential as a result of the mutation. The model analyses the impact of a variant on the splicing potential of residues in the surrounding genomic space.

SpliceAI consists of a 32-layer deep residual neural network, a subtype of neural network in which the network is arranged into so-called “residual blocks”—sub-networks containing “skip connections” that output directly to deeper layers in the model. This helps bypass common pitfalls for particularly deep neural networks, such as vanishing/exploding gradients, and also improves the speed with which the network learns [85].

To train the model, the authors selected over 20,287 principal protein-coding transcripts from the GENCODE v24 annotation, and used those from a selection of particular chromosomes (all except chr1, chr3, chr5, chr7, and chr9) as a training set, with the remainder acting as the test set, following removal of paralogs within the set. Each base within these transcripts was designated either a splice donor, acceptor or non-splice site. Four architectures were specifically designed: SpliceAI-80nt, SpliceAI-400nt, SpliceAI-2k, and SpliceAI-10k, where the suffix denotes the total number of bases flanking the variant that are input to the model.

SpliceAI is designed to infer features from the transcript sequence itself; as such, the only input to the model is a coded representation of the variant of interest and the flanking sequence of variable length, dependent on the above choice of model. Scores of gain or loss of acceptor or donor potential are generated for all residues lying within 50 bp of the variant on the pre-mRNA transcript. The residue within this flanking region that experiences the most significant change is then returned for each of these 4 consequences.

The authors demonstrate the ability of SpliceAI to faithfully identity true splice sites from nucleotide sequence alone, allowing recreation of entire gene transcripts; SpliceAI-10k exhibits 95% top-*k* accuracy and a PR-AUC (area under precision recall curve) of 0.98, both remarkably high figures. While the authors demonstrate very favorable model performance in comparison to earlier tools, e.g., MaxEntScan [53], GeneSplicer [86] and NNSplice [77], they did not analyze performance against any newer, machine-learning based tools. Such comparisons will prove very valuable in ascertaining the utility of SpliceAI in clinical practice.

In using a near-agnostic approach to model training, SpliceAI is able to identify features that may not be apparent to most humans. Because of this, it is quite possible that many features of the above tools, such as the modelling of competitive interactions between neighboring and novel splice sites, are already encompassed within the model. As acknowledged by the authors, however, this agnosticism may mean that certain features incorporated into the model do not truly reflect phenomena with biological meaning. Despite this, the power of the model, as well as the public availability of precomputed scores for all possible single nucleotide substitutions in the genome, suggest that SpliceAI may prove the gold standard for clinical interpretation of splice-impacting variants.

## 7. Discussion

The ever-growing range of splice prediction tools complicates variant interpretation by providing a surplus of choices for bioinformatics analysis. Identifying the optimal choice through direct, head-to-head comparisons of these tools is not a simple task. The genomic loci analyzable by different tools vary considerably, thus making construction of a universal test variant set difficult without the introduction of missing data points for at least one of the tools. The diverse functions of these tools also complicate comparative analysis. Comparing the performance of a tool predicting competitive splice site interactions with one predicting exon skipping, for example, may not ultimately prove informative.

Despite this, many of the papers describing the above tools do attempt such comparisons. SpliceAI, for instance, significantly outperforms the splice prediction tools GeneSplicer [86], MaxEntScan [53], and NNSplice [77] in both top-*k* accuracy and precision-recall. These latter tools are not machine learning-based, however, and were created over a decade ago, when sizeable training datasets were not so readily available. MMSplice [59], meanwhile, shows favorable performance over the similar tool COSSMO [87], and S-CAP [56] outperforms SPANR [54], CADD [61], TraP [62], and others across all six of its considered regions. Many tools described here are too novel to have conducted comparisons with one another; however, one recent comparison between the ability of three of these tools (SpliceAI, SPIDEX, and CADD) to correctly ascribe splice-impacting activity to variants showed SpliceAI to be superior in both sensitivity and specificity [88].

The different approaches adopted by these models offers clinical geneticists the opportunity to consider variant impact from many perspectives, both in terms of the specific splicing consequences predicted by the given model and the value it outputs. Broadly, tools may predict either pathogenicity or splicing impact. Care may need to be taken with the former, as training of a pathogenicity score is reliant on human annotations of pathogenicity, such as through ClinVar [57]. These annotations may be flawed, and may also suffer from ascertainment bias, whereby the main body of pathogenic variants in the database reflect the current state of our understanding of splice-impacting variants, thus underpowering models in the analysis of more elusive splice variant types. The ACMG have produced detailed guidelines for the scoring of variant pathogenicity [26]; consideration of splicing impact first and then following these guidelines on a variant-by-variant basis may prove a more robust and sensitive way to characterize pathogenic variants.

Machine learning models are often seen as “black boxes”, in that the inner workings of the model are not discernible to the user, and it is thus difficult for meaningful biological inferences to be made. However, variants flagged by these tools may prove a valuable jumping-off point for research into the mechanisms underlying the inability of earlier tools to correctly predict certain variants.

One such mechanism is the existence of long-distance splicing interactions: SpliceAI has demonstrated that consideration of wider genomic context significantly improves model performance. Such an improvement likely reflects the interactions between *trans*-acting splicing complexes bound across the often-significant lengths of introns, as well as their respective *cis*-acting binding sites [89,90]. Thus, SpliceAI may provide a useful resource in the investigation of long-range determinants of splicing, and ultimately improve our understanding of splicing in both a healthy and pathogenic context.

Many of these tools share common caveats. Few tools, for example, are able to predict the splice impact of indels, with the exceptions of CADD, MMSplice, and SpliceAI. Future tools will certainly benefit from more thorough consideration of such variants, which may have a significant impact on ultimate transcript structure. Indels affecting the poly-pyrimidine tract (PPT), for example, are known to have significant effects on splicing that may be more marked than the effect of many PPT SNVs, as spacing between the PPT and 3′ splice site is crucial for correct assembly of the spliceosome [91,92].

It should also be noted that atypical splice sites (i.e., those not consisting of GT-AG dinucleotide pairs) comprise just 1% of the body of human introns [93], and so do not feature prevalently in training sets. Some tools, such as CryptSplice, actively exclude such introns from model training. Thus, many models may be underpowered to predict changes affecting these low-frequency sites. The effect of variants in AT-AC introns (also known as U12 introns), which are instead processed by the biochemically distinct “minor spliceosome” [94], may be particularly difficult to predict. While the relative occurrence of such introns is low, they nonetheless represent a possible source of pathogenic variants [95], with mutations affecting the U12 5′-splice sites of introns in the STK11 [96] and TRAPPC2 [97] genes being shown to cause Peutz–Jeghers syndrome and spondyloepiphyseal dysplasia tarda, respectively. Special care may need to be taken, therefore, when considering variants in the vicinity of splice sites for such introns.

A final valuable consideration for models is the inclusion of more personalized and patient-specific prediction of splicing. The single-variant functionality of most of the above tools, for example, neglects to consider the interactions between multiple variants in close (or even distant) genomic space. Studies in mice suggest such interactions between common SNPs (i.e., an individual’s genetic background) and rare variants may underlie phenomena such as incomplete penetrance and variable expressivity [98,99] in Mendelian disorders. Consideration of these common genomic variants in tandem with variants of interest may allow further clarification of variants of uncertain significance.

The rapid increase in number of machine learning-based splice prediction tools, and the burgeoning power and efficacy of such tools, is an exciting development in the area of variant interpretation. However, it should be borne in mind that even the most powerful tools should not be taken as proof in isolation: the ACMG guidelines [26] assign only the lowest level of pathogenicity support to bioinformatics tools. All positive findings from splice prediction require corroboration using in vitro approaches, such as minigene splicing assays [100]. Thus, it is the combined consideration of predictive bioinformatics and functional analysis that will lead to the greatest strides forward in improving diagnostic yield for Mendelian disease patients.

## Figures and Tables

**Figure 1 cells-08-01513-f001:**
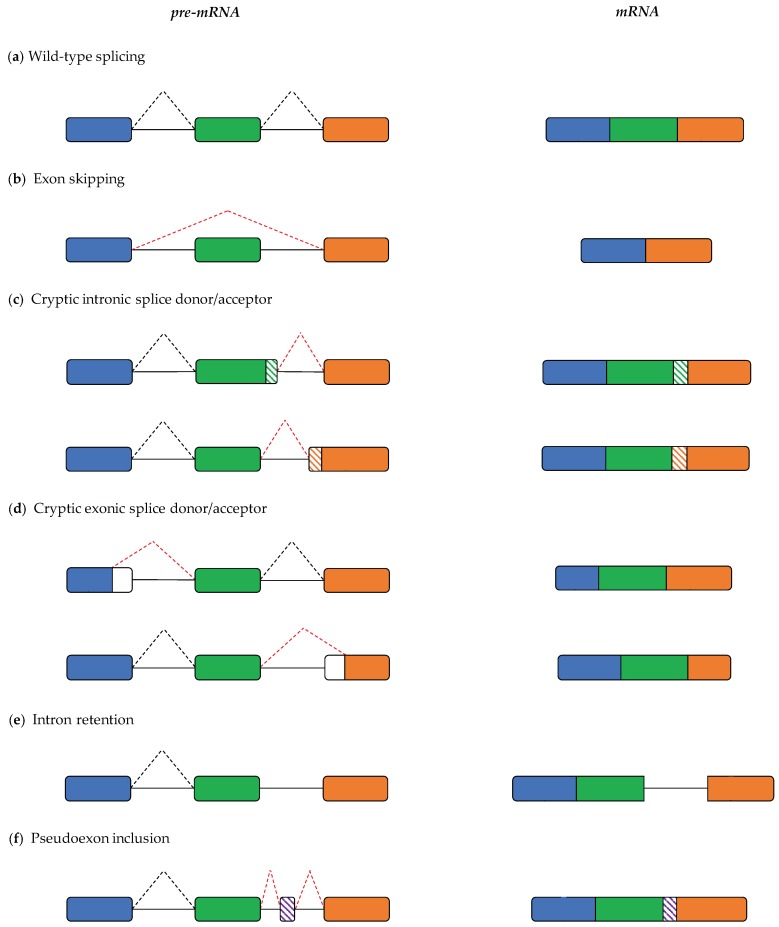
Diverse mechanisms of splicing dysfunction may be pathogenic. (**a**) Wild-type splicing. Schematic of a three-exon region of a gene (exons in blue, green and orange) with corresponding wild-type splicing activity. (**b**) Exon skipping. Mutations in or around an exon may lead to it being skipped from a final transcript. (**c**) Cryptic intronic splice donor/acceptor. Mutations in the intron may lead to generation of cryptic splice sites that outcompete canonical sites, leading to inclusion of intronic sequences. (**d**) Cryptic exonic splice donor/acceptor. Exonic mutations that activate cryptic sites may also outcompete canonical sites, causing exclusion of exonic sequences. (**e**) Intron retention. Splicing of a particular intron may be abrogated, leading to complete inclusion of the length of an intron. (**f**) Pseudoexon inclusion. Deeply intronic mutations may activate cryptic sites that aberrantly define lengths of intron as exonic, leading to inclusion of short segments of intronic sequence (pseudoexons). Solid lines, introns; black dashed lines, wild-type splicing; red dashed lines, mis-splicing events; hashed boxes, intronic regions aberrantly included as a result of a mis-splicing event; empty boxes, exonic regions that are usually retained after splicing, but which are erroneously excluded from the final transcript.

**Figure 2 cells-08-01513-f002:**
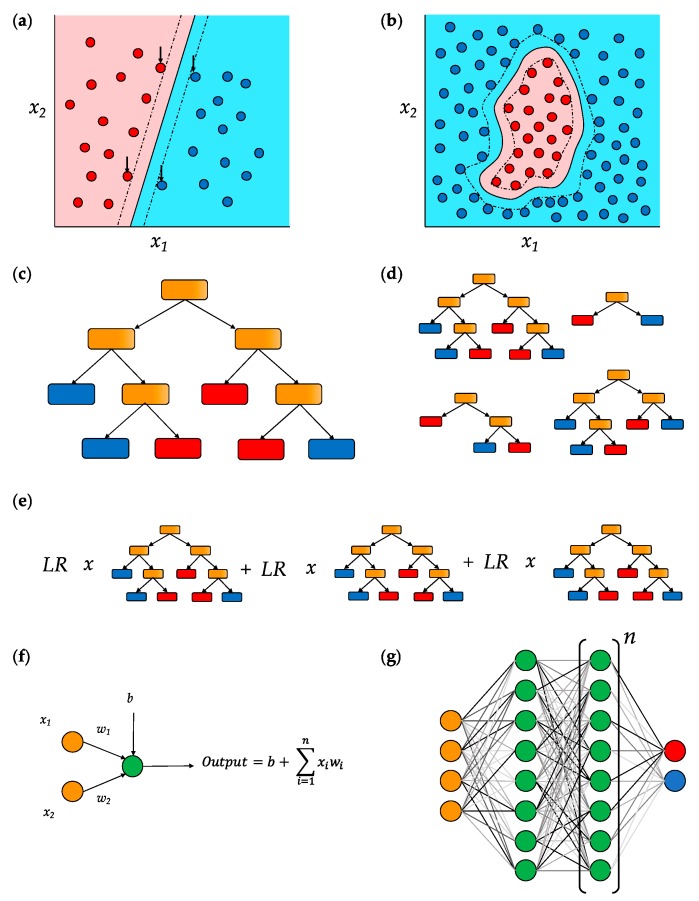
Basic machine learning models. (**a**) Support vector machines (SVMs) classify linearly-separable data using a single hyperplane (solid line), with points classified according to the side of the hyperplane on which they lie. Construction of the hyperplane is done using support vectors (indicated by arrows), data points that mark boundaries (dotted) within which the hyperplane must lie. (**b**) Where data are not linearly separable, they may be transformed using kernel functions (radial basis function, or RBF, kernel shown here) which infer relational qualities of data in a computationally inexpensive manner. (**c**) Decision trees use a series of binary choices (orange) to most effectively separate data into different categories (red and blue). (**d**) Random forest models consist of large numbers (often hundreds or thousands) of trees each derived from bootstrap aggregating (bagging) of both input features and data entries in the original training set. (**e**) To mitigate overfitting problems common to decision trees, gradient tree boosting generates successive trees of fixed structure that each contribute a small amount to the final classification, with each tree scaled by a *learning rate* between 0–1. (**f**) In a neural network, a single neuron receives quantitative input *(x*_i_*)* from neurons in the preceding layer and scales them according to the weights of its connection to them *(w*_i_*)*. Each neuron also has a “bias” *(b)*, representing a tendency for inactivity. The output (or *activation*) of a neuron is the sum of each input neuron multiplied by its respective weight, plus this bias value. (**g**) A deep neural network has an initial layer of input neurons (orange), which are coded representations of data features. These are connected to 1 or more layers of “hidden neurons” (green), which are, in turn, connected to an output layer of neurons (red and blue) corresponding to the possible classifications of the data. Predictions may be categorical or continuous and are based on the relative activation of the output neurons. Biases for each neuron and weights for each connection are randomized before the network is trained. After a set of training data is presented, the loss function of the model is calculated (i.e., how accurately or inaccurately the model has classified the known data) and an approach termed *backpropagation* is used to modulate each weight and bias so as to reduce this loss. More data is then presented and this process repeated iteratively to refine the model.

**Figure 3 cells-08-01513-f003:**
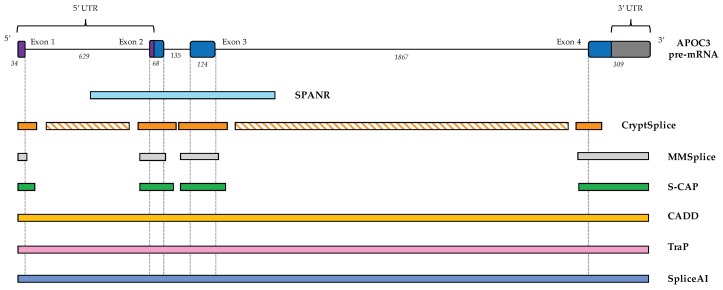
Location of variants amenable to analysis by splice prediction software. With diverse underlying training sets and purposes, different splice prediction tools are only able to analyze variants at particular sites in a pre-mRNA transcript. To-scale representation of the loci amenable to analysis by each of 7 tools for the pre-mRNA transcript of the human *APOC3* gene (RefSeq accession NM_000040.3). Dotted lines signify canonical exon-intron boundaries. Hashed bars represent loci where the variant effect can be modeled only if a novel splice donor or acceptor is created. Italicized numbers show exon/intron length in nucleotides. UTR, untranslated region.

**Figure 4 cells-08-01513-f004:**
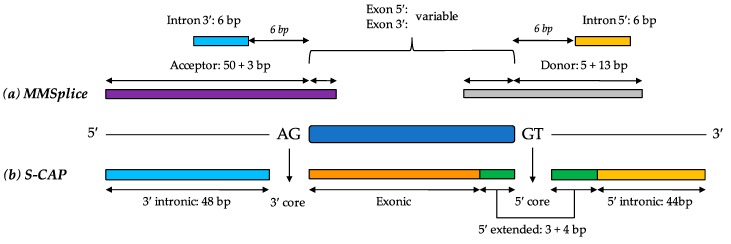
Compartmentalization of the splice region by S-CAP and MMSplice. Both MMSplice and S-CAP divide the splice region into six sub-regions, although the length and location of these divisions are different between the two tools. MMSplice (**a**) consists of 6 initial deep neural network modules corresponding to each region, with exonic and intronic modules both trained on the results of a massively parallel reporter assay (MPRA) experiment [64] and the acceptor and donor modules trained to predict functional acceptors and donors based on the real and decoy sites in the GENCODE v24 annotation. The scores from all modules are then passed to linear and logistic regression models to predict downstream effects, such as exon skipping, alteration of splicing efficiency, and competitive splice site interactions. S-CAP (**b**) consists of six separate models trained on pathogenic and benign variants curated for each region. The most significant consequence is returned for a given variant. Length of bars not to scale.

**Table 1 cells-08-01513-t001:** Glossary of machine learning terms. SVM, support vector machine.

Term	Definition
Backpropagation	The computational process by which a neural network adjusts the weights and biases of the network in such a way as to reduce the loss of the model.
Bagging	Abbreviation for bootstrap aggregation. The training of a model on random subsets of data entries and features to improve generalizability of a model (usually a decision tree-based model).
Bias	A (usually negative) value that represents a neuron’s inherent tendency towards inactivity. Usually randomized for each neuron before the training of a network.
Classification	A type of machine learning system in which the output is assignment of a data point to a discrete group. Usually contrasted with regression.
Feature	One of a set of variables in a dataset that are input to a machine learning model. Machine learning models classify data according to the values of features in the dataset.
Hidden layer	One of any number of layers of neurons lying between the input and output layers of a deep neural network.
Hyperplane	A surface with one fewer dimensions than the space it occupies. SVMs separate datasets with *n* features using a hyperplane of *n* − 1 dimensions. For example, if there are 6 features, an SVM attempts to create a 5-dimensional hyperplane that best separates data.
Kernel trick	The use of a mathematical function allowing inference of relational qualities of data without explicitly carrying out computationally expensive mathematical calculations.
Loss function	A mathematical function measuring the degree to which a model’s predictions deviate from the true classifications of data.
Machine learning	The use of computer systems to detect patterns in and make inferences from data without explicit instruction.
Multiclass SVM	A subtype of SVM used when data may be classified into more than two classes.
Neuron	The basic unit of a neural network, taking in input from previous neurons and propagating a weighted response to subsequent ones.
Regression	A type of machine learning system in which the output is the prediction of a continuous or ordered value. Usually contrasted with classification.
Support vectors	Data points that lie along the margins between classifications in an SVM model.
Training set	A dataset containing the data that is presented to a machine learning system and then used to make inferences and learn patterns present within the data.
Test set	The dataset used to evaluate performance of the model. The test set is generally taken from the same source as the training set, but may come from elsewhere.

**Table 2 cells-08-01513-t002:** Summary of splice prediction bioinformatics tools. Citation column denotes references to articles describing tools themselves. SVM, support vector machine; RBF, radial basis function; MPRA, massively parallel reporter assay; HGMD, Human Gene Mutation Database; PSSM, position-specific scoring matrix; pLI, probability of loss-of-function intolerant; RVIS, residual variation intolerance score; AUC, area under receiver operator characteristic (ROC) curve; PR-AUC, area under precision-recall curve.

Tool Name	Function	ML Model	Training/Testing Data	Features	Efficacy	Citation
CADD	General purpose pathogenicity scoring	v1.0: linear SVM Later releases: L_2_-regularized logistic regression	Benign training: evolutionarily neutral variants; pathogenic training: simulated de novo pathogenic variants Benign testing: common benign variants; pathogenic testing: pathogenic ClinVar variants, somatic cancer mutation frequencies	60, covering conversation scores, epigenetic modifications, functional analyses, and genetic context	AUC = 0.916, across all variant types	[55,61]
TraP	Quantification of variant impact on transcripts	Random forest of 1000 individual decision trees	Benign: De novo mutations in healthy individuals Pathogenic: Curated pathogenic synonymous variants	20, including several PSSM-based splice site scores, GERP++ conservation scores, and models of feature interactions	AUC = 0.88, all ClinVar variants AUC = 0.83, ClinVar intronic variants only	[62]
SPANR	Cassette exon skipping prediction	Group of neural networks modeled on Bayesian framework	ψ values for all human exons across 16 tissues, based on the Illumina Human Body Map project	1393, including exon/intron lengths, distances to nearest alternative splice sites, conservation and RNA secondary structure	AUC = 0.955, when distinguishing between high (≥67%) and low (≤33%) ψ values	[54]
CryptSplice	Effect of variants on existing splice sites and cryptic splice site prediction	SVM with RBF kernel	True and false splice sites from GenBank-derived datasets	3 types, all sequence-based, relating to the probability of finding given nucleotide sequences at certain points in splice region	Sensitivity = 97.8% and 88.9% in correctly labeling canonical donors and acceptors, respectively	[63]
MMSplice	Prediction of exon skipping, competitive interactions, changes in splicing efficiency and pathogenicity	Modular neural networks, and linear and logistic regression	Donor/acceptor modules: GENCODE v24 true and false splice sites Exon/intron modules: MPRA data from [64] Downstream models: various	Direct encoding of the sequence	R = 0.87 and 0.81, correlation between predicted and actual Δψ values for acceptor and donor mutations, respectively PR-AUC = 0.41, exon skipping prediction	[59]
S-CAP	Variant pathogenicity scoring with the compartmentalization of genomic space	Gradient boosting tree	Pathogenic variants curated from HGMD and ClinVar; benign variants curated from gnomAD	Features across chromosomal, gene, exon and variant levels, e.g., pLI [65], RVIS [66], CADD and SPIDEX scores, exon length, splice site strengths	AUC: 0.828–0.959, across 6 regions	[56]
SpliceAI	Prediction of variant impact on acceptor/donor loss or gain	32-layer deep neural network	GENCODE v24 pre-mRNA transcript sequence for human protein-coding genes	Direct encoding of the sequence	PR-AUC = 0.98 in correct prediction of splice site location from raw sequence	[60]

**Table 3 cells-08-01513-t003:** Tabulated list of loci on pre-mRNA transcripts amenable to predictive analysis by each of 7 splice prediction tools.

Tool	Loci Covered
SPANR	Any exon that is not first or terminal, plus 300 bp flanking intronic sequence
CryptSplice	Within 60 bp of a canonical splice junction; >100 bp into intron if novel donor/acceptor is created
MMSplice	Any exon, plus 50 bp upstream or 13 bp downstream
S-CAP	Any exon, plus 50 bp flanking intronic sequence
CADD	All loci
TraP	All loci
SpliceAI	All loci

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
