# Peer review of "Machine Learning Approaches for the Prioritization of Genomic Variants Impacting Pre-mRNA Splicing"

_cells, 2019, doi:10.3390/cells8121513_

Round 1

Reviewer 1 Report

cells-641046

Machine learning approaches for the prioritization of genomic variants impacting pre-mRNA splicing

Rowlands et al.

Splicing defects often result in genetic diseases in human. Prediction and analyses of abnormal splicing is critical to find cures for them. In this Review, the authors describe some basic principles of machine learning are introduced in the context of genomics and splicing analysis. They find that there is still much scope for personalized approaches to prediction of variant impact on splicing despite of great progress in producing specific and sensitive tools. Such approaches may increase diagnostic yields and underpin improvements to patient care.

This is an excellent and timely review for splicing analyses with splicing diseases. This review will be appreciated not only by informatics people but also RNA scientists.

I have one minor comment.

It would be nicer if the authors could compare different machine learning programs for prediction of the novel mutations mostly reported and its abnormal splicing resulting in to show the difference among them.

Author Response

Dear Editor at Cells,

Please find submitted a revised version of our Review article (cells-641046).

Many thanks to the editor and reviewers for your time and kind comments about our recent submission to Cells. As a result of your constructive feedback, we have made some changes to the manuscript.

We would be grateful if you would reconsider the revised manuscript. For ease, we have attached a point-by-point response to the points raised by the reviewers.

We look forward to your response.

Yours faithfully,

Charlie F Rowlands, Diana Baralle and Jamie M. Ellingford

Comment regarding direct comparison of machine learning tools

“It would be nicer if the authors could compare different machine learning programs for prediction of the novel mutations mostly reported and its abnormal splicing resulting in to show the difference among them.” (Reviewer 1)

This is a very valid and important issue, and one that is difficult to address for many of the reasons detailed in the opening paragraphs of the discussion section. We feel that conducting a direct comparison is beyond the remit of the review here, but we have added a new citation to point readers to recent work we have conducted that compares the performances of three of the tools we have described (Ellingford et al., 2019; citation 88 in the revised manuscript).

Comment regarding page numbering

“The numbering of the article is incorrect and on the last page (which should be 24 of 24), it appears as 14 of 24 and seems part of the guideline for the authors.” (Reviewer 2)

Many thanks for noticing this; we have now corrected the error in page numbering, which was due to the inclusion of a section break to accommodate the landscape table figure.

Comment regarding summary table

“Then, I miss a summary table for the methods. A table including the name of the method, a minimum description, highlighted features, the PMID and the download link. I think it would be very useful” (Reviewer 2)

While we did include a summary table covering basic information about each of the seven tools (page 9 of the revised manuscript), it lacked information about downloading or accessing the tools. We agree that this would be useful, and so we have added a table detailing the links for data access to the manuscript, along with PMIDs, in Appendix A, as Table A1.

And finally, could we please change the spelling of C. Rowlands author name. It should read Charlie F Rowlands.

Your Sincerely,

Prof Diana Baralle

For and on behalf of the authors.

Reviewer 2 Report

I think the review is wonderfully written. It introduce the reader to the necessary terms and purpose of the manuscript. Most of the article talks about the most common prediction methods using ML approaches and at the end there is a comparison/discussion among them.

I simply have two minor comments although the first one I doubt is due to the authors. The numbering of the article is incorrect and on the last page (which should be 24 of 24), it appears as 14 of 24 and seems part of the guideline for the authors.

Then, I miss a summary table for the methods. A table including the name of the method, a minimum description, highlighted features , the PMID and the download link. I think it would be very useful

Author Response

(The authors gave the same response as above.)
